# Impact of Impulsivity, Hyperactivity, and Inattention on Discontinuation Rate among Opioid-Dependent Patients Treated with Extended-Release Naltrexone

**DOI:** 10.3390/ijerph191811435

**Published:** 2022-09-11

**Authors:** Ann Tarja Karlsson, John-Kåre Vederhus, Thomas Clausen, Bente Weimand, Kristin Klemmetsby Solli, Lars Tanum

**Affiliations:** 1Addiction Unit, Sørlandet Hospital HF, 4604 Kristiansand, Norway; 2Norwegian Centre for Addiction Research, University of Oslo, 0315 Oslo, Norway; 3Center for Mental Health and Substance Abuse, University of South-Eastern Norway, 3040 Drammen, Norway; 4Department of R&D in Psychiatric Health Care, Akershus University Hospital, 1478 Oslo, Norway; 5Vestfold Hospital Trust, 3103 Tønsberg, Norway; 6Faculty for Health Science, Oslo Metropolitan University, 0130 Oslo, Norway

**Keywords:** extended-release naltrexone, retention in treatment, impulsivity

## Abstract

Previous studies have indicated elevated levels of impulsivity, hyperactivity, and inattention (IHI) among opioid-dependent patients seeking outpatient treatment with extended-release naltrexone (XR-NTX). This led us to hypothesize that IHI may be associated with a higher discontinuation rate for XR-NTX treatment. In a group of 162 patients with opioid dependence, discontinuation prior to the full 24 weeks of the study period (six injections and attending the study visit at 24 weeks) occurred in 49% of the patients, primarily in the early stage of treatment. IHI above the clinical cut-off on the adult ADHD self-report scale (ASRS) was not associated with a risk of premature discontinuation. This finding was not altered when controlling for socio-demographics, substance, use and mental health severity. Conclusively, high levels of IHI per se is not contradictive for XR-NTX treatment in regard to concern for premature discontinuation.

## 1. Introduction

Opioid use disorder (OUD) is considered a chronic condition, and the risk of relapse and overdose following detoxification and reversal of opioid tolerance is high [1]. The standard pharmacological treatment is agonist medication, such as buprenorphine (BUP), buprenorphine/naloxone (BUP-NX), or methadone, which is typically referred to as opioid maintenance treatment (OMT) [2]. Several types of medication are now also offered in extended-release suspensions, including buprenorphine (XR-BUP) [3]. Extended-release suspensions eliminate some of the known concerns about OMT, such as non-medical injection use of BUP and diversion of medication.

In the Norwegian NaltRec study, an extended-release formulation of the opioid antagonist naltrexone (NTX) was given as an alternative treatment option in OMT. In contrast to agonist medication, which covers the body’s need for opioids to the extent where physical withdrawal is kept in check, opioid antagonist treatment blocks opioid receptors in the brain, thereby eliminating the effect of exogenous opioids [4]. Patients often experience contemporary relief from the subjective experience of opioid cravings during treatment with NTX [5]. However, the sudden extinction of an often year-long need for opioids and the daily experience of a “fix-it-pill” can create an emotional vacuum for some patients. Other patients may not experience this relief from opioid cravings, or they may simply not be ready for life in abstinence for various reasons [6,7]. These scenarios could potentially lead to some patients prematurely discontinuing treatment with NTX. Knowing that abstinence over time will reverse tolerance and, therefore, increase the risk of overdose if opioid drugs are taken, makes the discontinuation of NTX a dangerous act if patients relapse into high-risk opioid use [8]. This is explicitly the case if relapsing into illicit opioids, such as heroin or fentanyl, since dosing and administration is uncontrolled compared to the conditions of agonist medication in OMT. When studying the treatment of substance use disorders (SUDs), a multitude of factors may influence prognosis and retention. In addition to the more obvious variables, such as demographics and severity of the SUD, a previous Norwegian study found that enrollment in OMT prior to entering a study on extended-release NTX (XR-NTX) was significantly associated with retention in treatment [9].

There are several known risk factors for the discontinuation of treatment in the field of SUDs, including cognitive dysfunction [10]. Previously, we found that four out of 10 patients have a score on the Adult ADHD Self-report Scale (ASRS) above the clinical cut-off, indicating high levels of self-reported impulsivity, hyperactivity, and inattention (IHI) [11]. Depending on severity, these traits can constitute impairment in executive functioning. This led us to hypothesize that these patients may be more likely to discontinue treatment prematurely than patients with ASRS scores below the cut-off. If this hypothesis is supported, clinicians may better accommodate the potential additional needs of these patients, such as in the form of interventions tailored to prevent the discontinuation of treatment due to a sudden impulse when the next injection is due.

We investigated the treatment discontinuation rate in a population of opioid-dependent patients qualifying for OMT medication, with a preference for a 24-week treatment period with XR-NTX. We aimed to uncover whether impulsive patients (i.e., with ASRS scores above the clinical cut-off) were more likely to end treatment than patients scoring below the cut-off. For this analysis, we also controlled for other factors relevant to discontinuation.

## 2. Materials and Methods

Data in this article were collected in the Norwegian NaltRec study (“Long acting naltrexone for opioid addiction: the importance of mental, physical and societal factors for sustained abstinence and recovery”) [12]. Five Norwegian hospitals participated in this phase 4 open label prospective multi-center study: Akershus University Hospital, Sørlandet Hospital, Vestfold Hospital Trust, Oslo University Hospital, and Haukeland University Hospital. XR-NTX was studied in the treatment and recovery process of patients with opioid dependence qualifying for OMT in a naturalistic setting.

Following a complete tapering of opioids, patients were inducted on an intramuscular suspension of 380 mg of extended-release naltrexone hydrochloride (Vivitrol^®^) releasing therapeutic dosages over a period of approximately 28 days. The study was construed with a 24-week study period, including 6 injections of XR-NTX and attending the study visit at 24 weeks, where patients could opt to participate in an additional 28-week (7 injection) treatment. Here, we focus on the first 24-week study period. At each study visit, data were collected in the form of study interviews and assessments of mental distress.

The study was carried out in accordance with the guidelines for Good Clinical Practice (GCP) following the ethical principles originating from the Declaration of Helsinki [13] and consistent with the International Conference on Harmonization (ICH) as well as national regulatory requirements. The General Data Protection Regulation (GDPR) and the National Personal Data Protection regulations were followed when registering and handling patient data.

Prior to the start of the study, all patients provided informed consent and were either enrolled or continued in an OMT program at inclusion, ensuring immediate follow-up, counselling and availability of opioid agonist therapy if needed due to the potential discontinuation of XR-NTX. Representatives from user organizations were involved in developing the study.

### 2.1. Setting and Patients

Study enrollment occurred in Norway during the period between September 2018 and September 2020, and patients were mainly recruited from addiction clinics and detoxification units at the participating hospitals. Other recruitment channels were municipal health services and through the network of already recruited patients. Inclusion criteria were a current diagnosis of opioid dependence [14] and capability and willingness to understand and comply with study procedures. Both sexes aged 18–65 years were eligible, and enrollment in an OMT program was mandatory. Fertile women were required to use safe contraception of their choice. Exclusion criteria were severe psychiatric or physical illness demanding treatment that could interfere with study participation, pregnancy and lactation, or concomitant alcohol dependence. In the present study, we included patients who gave consent and expressed an intention to start treatment with XR-NTX (*n* = 171) and completed the full ASRS at baseline (*n* = 162).

### 2.2. Measures

At each study visit, data were collected in the form of study interviews as described below. Trained staff at the participating hospitals carried out all screening procedures. The European version of the Addiction Severity Index (Europ-ASI) was used to collect demographic data and The Mini International Neuropsychiatric Interview, version 6.0, was used to confirm ongoing opioid dependence [14].

#### 2.2.1. Main Outcome

Discontinuation was the main outcome and defined as receiving <6 injections of XR-NTX due to failed attendance at a study visit and XR-NTX injection, unwillingness to comply with study procedures, or not wanting to proceed without giving further reason. Per protocol, patients were also required to meet at the 24-week study visit to avoid being considered a non-completer. We used the number of cumulated injections for each patient as a measure for time in the study (recommended time between each injection was 28 days).

#### 2.2.2. IHI

IHI was measured by the 18-item ASRS (ASRS-18). This scale has questions that target the patients’ own perception and evaluation of impulsivity/hyperactivity and inattention in everyday situations. With nine questions aimed at each area, patients were asked to rate the occurrence of one’s own symptoms of IHI during the past 6 months on a 5-point Likert scale (0, never; 1, rarely; 2, sometimes; 3, often; 4, very often). Each question was dichotomized according to the description of Kessler et al. [15]. For 7 of the questions, clinical significance was set at a score of 2–4 (e.g., “How often do you have difficulty getting things in order when you have to do a task that requires organization?”), whereas a score of 3–4 (e.g., “How often do you have difficulty waiting your turn in situations when turn taking is required?”) was considered clinically significant for the remaining 11 questions. To create a clinical symptom level for IHI, a summed score was calculated and then dichotomized by dividing patients into below and above the cut-off, which was 9 as suggested by Kessler [15]. The ASRS scores were calculated post study interviews, blinding clinicians to patients’ IHI status. We use the term ASRS+ for patients scoring ≥9 and ASRS- for patients scoring <9 on the ASRS. 

#### 2.2.3. Mental Health

The 25-item Hopkin’s Symptom Checklist (H-SCL-25) was used to measure mental distress [16]. Self-reported symptoms of anxiety and depression during the past 14 days were plotted on a 4-point Likert scale ranging from 1 (“not at all bothered”) to 4 (“extremely bothered”). The Global Severity Index (GSI) refers to a mean score of all items, and the cut-off score for clinical levels of mental distress was set at 1.75.

We used the Europ-ASI to assess the severity of alcohol and drug use [17]. Severity was measured by calculating composite scores based on the past 28 days and converted into scores from 0 (no problem) to 1 (a severe problem) in each area.

### 2.3. Statistical Analysis

Descriptive statistics were used to characterize the sample. We used the chi-squared test and the log rank test to compare discontinuation between groups. A Cox regression analysis was used to control for relevant covariates (socio-demographic characteristics and severity variables). Patients who completed the treatment (6 injections and study visit at 24 weeks) or who developed a health condition incompatible with XR-NTX treatment were considered censored in the statistical analysis. The results of this analysis are presented as hazard ratios (HRs) with 95% confidence intervals (CIs). The threshold for significance was set at *p* < 0.05. All statistical analyses were performed using IBM SPSS, version 26 (IBM Corporation, Armonk, NY, USA).

## 3. Results

A total of 162 patients (mean age 38 years, SD = 10; 24% women) were included in this study (Table 1). At baseline, living conditions during the past 6 months were reported as unstable for 18 patients (11%). Prior to study participation, 61 patients (38%) had not previously been enrolled in any OMT program and 66 (41%) were categorized as ASRS+ (Table 1).

Among those who expressed a desire for XR-NTX, 27 (17%) never initiated the medication, with no significant difference between the ASRS+ and ASRS- groups. Of the patients inducted on XR-NTX, the share who discontinued treatment after the first injection was 15% (20 of 135), with no significant difference between the groups. Cumulatively, this means that 29% (47 of 162) were not retained in the medical treatment after 28 days. At the end of the 24-week study, 79 patients (49%) had discontinued treatment, with 50% in the ASRS+ group and 48% in the ASRS- group (χ^2^ = 0.07, *p* = 0.794). The discontinuation rate for the two groups is shown in the Kaplan–Meier curve in Figure 1. Considering the retention time trajectories, a log rank test showed no significant difference between the two groups after 24 weeks (χ^2^ = 0.03, df = 1, *p* = 0.859).

When controlling for socio-demographic factors, substance use, and mental health severity in the Cox regression model, the difference between groups remained insignificant (Table 2). None of the other co-variables included in the analysis made significant contributions to explaining the discontinuation of treatment.

## 4. Discussion

We found no significant difference between the ASRS+ and ASRS- groups regarding discontinuation of treatment in the 24-week study period. Socio-demographic factors, severity of substance use, and mental health were not significantly associated with premature discontinuation.

A review paper with pooled data published in 2018 found that adherence to XR-NTX treatment generally decreased over time [18]. Approximately 41% of patients in that study were still in treatment at the latest time point, which was less than 4 months on average. Our finding was slightly better, with roughly half of the patients retained after 6 months in treatment. Although a recent Norwegian study found that the retention rate for XR-NTX was non-inferior to BUP [4], a recent US study showed that among various types of medication for opioid use disorder, NTX had the highest rate of discontinuation [19,20]. In one US study, 52% of patients discontinued treatment after the initial injection of XR-NTX [19], whereas only 15% in our sample discontinued after the first injection. The studies may not be directly comparable because Morgan et al. conducted a retrospective study conducted in a different treatment setting [19]. Previous studies have shown that adherence is better in clinical prospective studies than retrospective studies, implying that the level of follow-up affects adherence to XR-NTX [18]. In addition, in the US study, those who did not initiate the medication (pre-study discontinuation) were not included in the analyses. If these patients had been included, the cumulative percentage would have been even higher than 52%. Despite the US study having a higher discontinuation rate in the very early stage of treatment, we still consider the 29% of patients intending to start treatment with XR-NTX but not continuing treatment after 28 days to be rather high. Included in this group are a fair number of patients (17%) who wish to receive treatment with XR-NTX and end up rejecting it before treatment is initiated. One of the known obstacles regarding induction is the detoxification process, when opioids are tapered and patients undergo physical withdrawal [21], but the concerns regarding a novel form of treatment may also affect induction [22]. This could offer a plausible explanation as to why some of our patients do not follow through with the induction of XR-NTX.

In our study, IHI did not push patients toward the premature discontinuation of XR-NTX. This contrasted with our main hypothesis regarding the possible elevated rate of premature discontinuation of treatment with XR-NTX with a subsequent risk of impulsive relapse into opioid use in patients with high levels of IHI. However, one cannot dismiss concerns regarding IHI based on our study results alone. The study procedures and interviews we carried out constituted an intervention similar to counseling and an amount of time spent with health care personnel that exceeds the average for the majority of patients in ordinary OMT. In our study, this condition is equal for all patients regardless of their levels of IHI. However, patients who may be difficult to retain in treatment due to impulsivity may also benefit from a more intensive follow-up. Furthermore, the research assistants were experienced clinicians, and these patients may already receive extra clinical attention adjusted to their condition, although clinicians were blinded to ASRS scores. Thus, the steps and interventions considered to be necessary therapeutic additions to accommodate the patient’s levels of IHI alongside medical treatment may already have been established unintentionally due to the clinical expertise and background of the study workers and staff in the OMT program.

We found that none of the investigated covariates (socio-demographic factors and severity variables) were associated with the discontinuation of XR-NTX. Other studies have found that young age correlates with discontinuation [10,23], but in line with our findings, most patient characteristics fail as clear predictors or have not been replicable across studies [18].

Apparently, the reasons for discontinuing treatment with XR-NTX are individual. An outpatient study carried out in the US revealed that a frequent reason for discontinuing XR-NTX was a feeling of “having been cured” or a wish to remain abstinent without medications [24]. Discontinuation may also reflect an underlying feeling of oppression in patients with a long history of OMT, as this is a highly regulated treatment form for safety reasons [25]. Other patients discontinuing treatment with a wish to remain abstinent may simply underestimate the struggle of being sober and risk of relapse over time. Although our study did not include data on relapse or substance use for those who chose to end treatment prematurely, it seems unlikely that these patients would be fully recovered given the known circumstances regarding the nature of OUD and the risk of relapse [26].

### Methodological Considerations

The measures for both IHI and mental distress rely on patients’ own perceptions of symptom load and, therefore, may suffer the risk of being influenced by their current emotional state. However, trained staff bear this in mind during screening procedures in order to eliminate the influence of ongoing withdrawal or substance use.

The situation with the global pandemic made it extra challenging to collect data and carry out a longitudinal medical study such as the present analysis. During periods of lock-down, patient access to hospital areas were limited to injection procedures only, leaving the study interviews to be carried out by phone. This may have influenced the experience of follow-up for some patients and affected the retention time in the study. Future studies should examine the reasons why patients discontinue XR-NTX to provide greater insight into treatment obstacles.

## 5. Conclusions

The discontinuation rate for XR-NTX treatment among opioid-dependent patients during a 24-week period was 49% in our study. Levels of IHI above the clinical cut-off did not imply a higher discontinuation rate. The scope of follow-up offered in addition to medical treatment may affect adherence; therefore, clinicians should intensify this parameter if patients are offered XR-NTX treatment.

## Figures and Tables

**Figure 1 ijerph-19-11435-f001:**
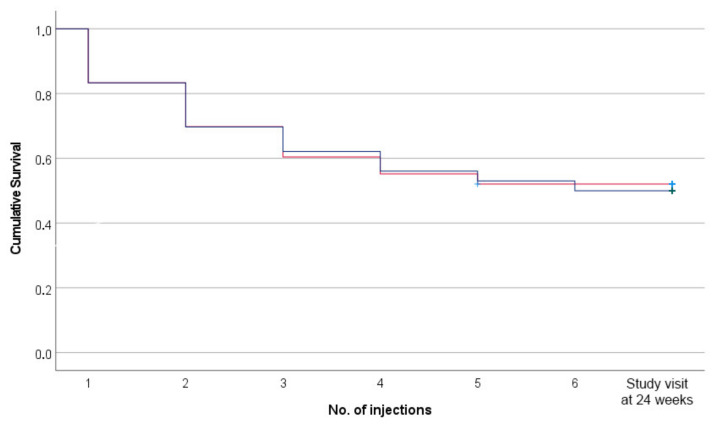
Kaplan–Meier curve of the discontinuation rate for XR-NTX. Red line = ASRS < 9 (*n* = 96). Blue line = ASRS ≥ 9 (*n* = 66). Log rank test = 0.03 (*p* = 0.859). Injections were given after detox and every 4 weeks. The last marker on the x-axis represents attending the 24-weeks study visit.

**Table 1 ijerph-19-11435-t001:** Socio-demographic and severity variables at baseline (*n* = 162).

Variable	ASRS < 9	ASRS ≥ 9 (*n* = 66)	Total (*n* = 162)
	*n* = 96	*n* = 66	*n* = 162
Female gender	21 (22%)	18 (27%)	39 (24%)
Age, years	38 (10)	37 (10)	38 (10)
Unstable living conditions past 6 months	11 (11%)	7 (11%)	18 (11%)
Years of completed education	12.2 (2.5)	11.4 (2.5)	11.9 (2.5)
Not in OMT before enrollment in study	41 (43%)	20 (30%)	61 (38%)
Severity variables at baseline:			
Alcohol use	0.04 (0.06)	0.08 (0.12)	0.06 (0.09)
Drug use	0.31 (0.16)	0.35 (0.19)	0.33 (0.17)
Mental distress (H-SCL-25 GSI)	1.79 (0.49)	2.16 (0.65)	1.94 (0.58)

Results are reported as *n* (%) or mean (SD). Unstable living conditions was based on Europ-ASI and defined as either no stable living condition or in prison/institution. Severity of alcohol and drug use was based on composite scores from Europ-ASI. ASRS = Adult ADHD Self-Report Scale 18-item version (ASRS-18), OMT = opioid maintenance treatment, H-SCL-25 GSI = Hopkin’s Symptom Checklist Global Severity Index.

**Table 2 ijerph-19-11435-t002:** Cox regression containing examined predictors for discontinuation of XR-NTX (*n* = 162).

Variable	Β	HR	CI	*p*-Value
ASRS > cut-off	−0.14	0.87	0.53–1.45	0.594
Gender	0.34	1.40	0.86–2.29	0.179
Age	0.00	1.00	0.98–1.02	0.955
Years of education	−0.06	0.94	0.85–1.05	0.267
Living conditions past 6 months	0.41	1.50	0.77–2.92	0.228
In OMT prior to study inclusion	0.33	1.39	0.82–2.36	0.219
Baseline severity alcohol use	−1.50	0.22	0.01–4.12	0.314
Baseline severity drug use	−0.34	0.71	0.17–3.01	0.646
Baseline SCL-GSI	0.30	1.35	0.88–2.06	0.165

Living conditions were based on Europ-ASI and defined as either no stable living condition or in prison/institution. Baseline severity of alcohol and drugs was based on composite scores in Europ-ASI. HR = hazard ratio, CI = confidence interval, ASRS = Adult ADHD Self-Report Scale, OMT = opioid maintenance treatment, SCL-GSI = Hopkin’s Symptom Checklist Global Severity Index.

## Data Availability

The present article builds upon data from an ongoing study with an anticipated end date in 2025. Anonymized data will be available publicly in a suitable repository (e.g., The Norwegian Centre for Research Data) in accordance with current Norwegian regulation and practice.

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
