# Peer review of "Impact of Impulsivity, Hyperactivity, and Inattention on Discontinuation Rate among Opioid-Dependent Patients Treated with Extended-Release Naltrexone"

_ijerph, 2022, doi:10.3390/ijerph191811435_

Round 1

Reviewer 1 Report

The study described in this manuscript covers the question of whether impulsivity, hyperactivity and inattention might be related to discontinuation of extended-release naltrexone treatment for opioid dependent abstinent subjects. The topic is of interest and is supported by previous data and the experiments and analyses of the data were performed correctly. Notwithstanding the aforementioned, I would like to point out some details which raised me some questions.

- The title could be a little misleading and the reader could misjudge that IHI is indeed  related to higher levels of discontinuation of XR-NTX treatment.

Introduction:

-  There is not mention in the introduction of how XR-NTX treatment compares with other available treatments.

- Line 41: I think the authors meant "temporary" instead of "contemporary"

Materials & methods

- In the discussion, the authors mention that the patients with high IHI scores might have received extra clinical attention by the research assistants due to their condition, but it is not clear whether they were aware of their tests results or they were blind, but they might have perceived it due to their experience. The methods section should mention wether they where blind to the IHI status of the patients or not.

- I think it would have been interesting if the authors had checked the reasons for discontinuation of the treatment and compared the results among the IHI+ and IHI- groups.

- Line 105: "obligated" sounds to me as if they were forced to use contraception. I think "required" would sound less agressive.

Results:

- A flowchart of the clinical trial process would be appreciated. 

Discussion:

-  The following sentence seems very "out of place":

"Taking into consideration the personal costs and challenges being part of the lives of people suffering from opioid dependence, one would assume that these people would opt for and gain an advantage from any type of assistance available to become and remain abstinent. This assumption seems to be contradicted by the findings in our study, with as many as half the patients discontinuing treatment after 6 months."

- Line 210: delete "we"?

- Line 221: who wish to receive treatment with XR-NTX "and" end up rejecting

Author Response

See enclosed file. 

Reviewer 2 Report

Review of manuscript entitled: “Discontinuation of extended-release naltrexone in opioid-dependent patients with high levels of impulsivity, hyperactivity, and inattention” authored by Ann Tarja Karlsson, John-Kåre Vederhus, Thomas Clausen, Bente Weimand, Kristin Klemmetsby Solli and Lars Tanum.

Presented manuscript undertakes very important problem associated with opioid use disorder and its treatment. The article is well-written and provides new information about the connection between IHI (impulsivity, hyperactivity and inattention) and discontinuation rate for XR-NTX treatment. I don’t have any major concerns about the quality of the manuscript.

Introduction is written in a proper way and provides sufficient background about undertaken problem.

Materials and methods section provides sufficient information about the experiment, however I have some questions:

·         Why such a dose of extended-release naltrexone was used and what was the daily release of the substance? Why the study time was 24-weeks?

Setting and patients: Were all of the patients included in the study Norwegian nationality? If so, maybe it would be good to add it to the title of the manuscript?

Mental health:

·         Did patients have access to psychiatric treatment, if needed?

·         Did the used treatment have any side effects reported by patients?

Author Response

See enclosed file. 

Reviewer 3 Report

Comments to the authors:

The present study, “Discontinuation of extended-release naltrexone in opioid-dependent patients with high levels of impulsivity, hyperactivity, and inattention (IHI),” examines whether discontinuing extended-release naltrexone (XR-NTX) was associated with high responses in impulsivity, hyperactivity, and inattention. The present result showed that the discontinuation of XR-NTX before the 24 weeks that was given six injections and attending the study visits had 49% of the patients. IHI’s score was higher than the cut-off score on the adult ADHD self-report scale and was not associated with a risk of premature discontinuation. This topic of the study is very interesting. However, the study used descriptive statistics and cox regression. Their results showed nonsignificant differences for all testing values such as ASRS-cut-off, gender, age, years of education, living conditions past six months, in OMT before study inclusion, baseline severity alcohol use, baseline severity drug use, and baseline SCL-GSI (all, p > 0.05) in Table 2. Table 1 depicts sociodemographic and severity variables at baseline. Seriously, Table 1 did not analyze collected data; instead, it just uses the descriptive statistics method to introduce their values. Figure 1 also gets the same problem. Figure 1 did not show the significant differences between the ASRS<9 and ASRS > 9 lines using Log-rank tests. Therefore, the nonsignificant differences in the study appeared to be a weak power for explaining the results. In summary, the present study was not considered for acceptance.

Author Response

See enclosed file

Round 2

Reviewer 2 Report

Authors responded to all of my questions and concerns. I find this answers sufficient.

Author Response

Reviewer 2:

Authors responded to all of my questions and concerns. I find this answers sufficient.

Response: Thank you for the positive feedback.

Reviewer 3 Report

Dear Editors,

Greetings for the day.

This manuscript has fundamental problems, and their results got nonsignificant differences. I do not think this study has good contributions to this field of drug addiction. The authors' responses did not fully answer my questions. This manuscript should not be accepted because it did not have any novel idea, and the data did not support its hypothesis. Also, the authors cannot provide an alternative viewpoint based on the present data. 

Best regards,

Author Response

Response: In the first review, the reviewer stated that the topic of the study is very interesting, but in the second review, s/he does not think that the study has good contributions to the field of drug addiction and finds the paper non-acceptable. We wonder if his/her suggestion to reject our manuscript is based on the fact that we had non-significant findings and that these surprisingly did not support our hypotheses. The reviewer’s response seems to indicate that

  1. Hypotheses should always be supported by the results
  2. Analyses should always yield significant findings

Unfortunately, we disagree on these viewpoints. In our previous response, we acknowledged that our main hypothesis was not supported. However, the discussion section outlines why the findings is of clinical value. To our knowledge, no previous study has investigated the impact of impulsivity, hyperactivity, and inattention (IHI) on discontinuation of XR-NTX. We believe that the study has merit in the field of drug addiction since it is, at least in part, contradictive to perceived beliefs about IHI and discontinuation of treatment. As we would have answered any questions from reviewer 3 there were no specific questions stated. Further, we do not believe that it is of good research ethics to formulate new hypothesis based on a dataset with the intention of supporting this with significant findings.